# Coin Betting and Parameter-Free Online Learning

**Francesco Orabona**
Stony Brook University, Stony Brook, NY
francesco@orabona.com

**Dávid Pál**
Yahoo Research, New York, NY
dpal@yahoo-inc.com

## Abstract

In the recent years, a number of parameter-free algorithms have been developed for online linear optimization over Hilbert spaces and for learning with expert advice. These algorithms achieve optimal regret bounds that depend on the unknown competitors, without having to tune the learning rates with oracle choices.

We present a new intuitive framework to design parameter-free algorithms for *both* online linear optimization over Hilbert spaces and for learning with expert advice, based on reductions to betting on outcomes of adversarial coins. We instantiate it using a betting algorithm based on the Krichevsky-Trofimov estimator. The resulting algorithms are simple, with no parameters to be tuned, and they improve or match previous results in terms of regret guarantee and per-round complexity.

## 1 Introduction

We consider the Online Linear Optimization (OLO) [4, 25] setting. In each round $t$, an algorithm chooses a point $w_t$ from a convex *decision set* $K$ and then receives a reward vector $g_t$. The algorithm's goal is to keep its *regret* small, defined as the difference between its cumulative reward and the cumulative reward of a fixed strategy $u \in K$, that is

$$\text{Regret}_T(u) = \sum_{t=1}^{T} \langle g_t, u \rangle - \sum_{t=1}^{T} \langle g_t, w_t \rangle \ .$$

We focus on two particular decision sets, the $N$-dimensional probability simplex $\Delta_N = \{x \in \mathbb{R}^N : x \geq 0, \|x\|_1 = 1\}$ and a Hilbert space $\mathcal{H}$. OLO over $\Delta_N$ is referred to as the problem of Learning with Expert Advice (LEA). We assume bounds on the norms of the reward vectors: For OLO over $\mathcal{H}$, we assume that $\|g_t\| \leq 1$, and for LEA we assume that $g_t \in [0,1]^N$.

OLO is a basic building block of many machine learning problems. For example, Online Convex Optimization (OCO), the problem analogous to OLO where $\langle g_t, u \rangle$ is generalized to an arbitrary convex function $\ell_t(u)$, is solved through a reduction to OLO [25]. LEA [17, 27, 5] provides a way of combining classifiers and it is at the heart of boosting [12]. Batch and stochastic convex optimization can also be solved through a reduction to OLO [25].

To achieve optimal regret, most of the existing online algorithms require the user to set the learning rate (step size) $\eta$ to an unknown/oracle value. For example, to obtain the optimal bound for Online Gradient Descent (OGD), the learning rate has to be set with the knowledge of the norm of the competitor $u$, $\|u\|$; second entry in Table 1. Likewise, the optimal learning rate for Hedge depends on the KL divergence between the prior weighting $\pi$ and the unknown competitor $u$, $\text{D}(u\|\pi)$; seventh entry in Table 1. Recently, new parameter-free algorithms have been proposed, both for LEA [6, 8, 18, 19, 15, 11] and for OLO/OCO over Hilbert spaces [26, 23, 21, 22, 24]. These algorithms adapt to the number of experts and to the norm of the optimal predictor, respectively, without the need to tune parameters. However, their *design and underlying intuition* is still a challenge. Foster et al. [11] proposed a unified framework, but it is not constructive. Furthermore, all existing algorithms for LEA either have sub-optimal regret bound (e.g. extra $\mathcal{O}(\log \log T)$ factor) or sub-optimal running time (e.g. requiring solving a numerical problem in every round, or with extra factors); see Table 1.

| Algorithm | Worst-case regret guarantee | Per-round time complexity | Adaptive | Unified analysis |
|---|---|---|---|---|
| OGD, $\eta = \frac{1}{\sqrt{T}}$ [25] | $\mathcal{O}((1+\|u\|^2)\sqrt{T}), \forall u \in \mathcal{H}$ | $\mathcal{O}(1)$ | | |
| OGD, $\eta = \frac{U}{\sqrt{T}}$ [25] | $U\sqrt{T}$ for any $u \in \mathcal{H}$ s.t. $\|u\| \leq U$ | $\mathcal{O}(1)$ | | |
| [23] | $\mathcal{O}(\|u\|\ln(1+\|u\|T)\sqrt{T}), \forall u \in \mathcal{H}$ | $\mathcal{O}(1)$ | ✓ | |
| [22, 24] | $\mathcal{O}(\|u\|\sqrt{T\ln(1+\|u\|T)}), \forall u \in \mathcal{H}$ | $\mathcal{O}(1)$ | ✓ | |
| This paper, Sec. 7.1 | $\mathcal{O}(\|u\|\sqrt{T\ln(1+\|u\|T)}), \forall u \in \mathcal{H}$ | $\mathcal{O}(1)$ | ✓ | ✓ |
| Hedge, $\eta = \sqrt{\frac{\ln N}{T}}, \pi_i = \frac{1}{N}$ [12] | $\mathcal{O}(\sqrt{T\ln N}), \forall u \in \Delta_N$ | $\mathcal{O}(N)$ | | |
| Hedge, $\eta = \frac{U}{\sqrt{T}}$ [12] | $\mathcal{O}(U\sqrt{T})$ for any $u \in \Delta_N$ s.t. $\sqrt{D(u\|\pi)} \leq U$ | $\mathcal{O}(N)$ | | |
| [6] | $\mathcal{O}(\sqrt{T(1+D(u\|\pi))}+\ln^2 N), \forall u \in \Delta_N$ | $\mathcal{O}(NK)^1$ | ✓ | |
| [8] | $\mathcal{O}(\sqrt{T(1+D(u\|\pi))}), \forall u \in \Delta_N$ | $\mathcal{O}(NK)^1$ | ✓ | |
| [8, 19, 15][2] | $\mathcal{O}(\sqrt{T(\ln\ln T + D(u\|\pi))}), \forall u \in \Delta_N$ | $\mathcal{O}(N)$ | ✓ | |
| [11] | $\mathcal{O}(\sqrt{T(1+D(u\|\pi))}), \forall u \in \Delta_N$ | $\mathcal{O}(N\ln\max_{u\in\Delta_N} D(u\|\pi))^3$ | ✓ | ✓ |
| This paper, Sec. 7.2 | $\mathcal{O}(\sqrt{T(1+D(u\|\pi))}), \forall u \in \Delta_N$ | $\mathcal{O}(N)$ | ✓ | ✓ |

Table 1: Algorithms for OLO over Hilbert space and LEA.

**Contributions.** We show that a more fundamental notion subsumes *both* OLO and LEA parameter-free algorithms. We prove that the ability to maximize the wealth in bets on the outcomes of coin flips *implies* OLO and LEA parameter-free algorithms. We develop a novel potential-based framework for betting algorithms. It gives intuition to previous constructions and, instantiated with the Krichevsky-Trofimov estimator, provides new and elegant algorithms for OLO and LEA. The new algorithms also have optimal worst-case guarantees on regret and time complexity; see Table 1.

## 2 Preliminaries

We begin by providing some definitions. The Kullback-Leibler (KL) divergence between two discrete distributions $p$ and $q$ is $D(p\|q) = \sum_i p_i \ln(p_i/q_i)$. If $p, q$ are real numbers in $[0, 1]$, we denote by $D(p\|q) = p\ln(p/q) + (1-p)\ln((1-p)/(1-q))$ the KL divergence between two Bernoulli distributions with parameters $p$ and $q$. We denote by $\mathcal{H}$ a Hilbert space, by $\langle\cdot,\cdot\rangle$ its inner product, and by $\|\cdot\|$ the induced norm. We denote by $\|\cdot\|_1$ the 1-norm in $\mathbb{R}^N$. A function $F : I \to \mathbb{R}_+$ is called *logarithmically convex* iff $f(x) = \ln(F(x))$ is convex. Let $f : V \to \mathbb{R} \cup \{\pm\infty\}$, the Fenchel conjugate of $f$ is $f^* : V^* \to \mathbb{R} \cup \{\pm\infty\}$ defined on the dual vector space $V^*$ by $f^*(\theta) = \sup_{x\in V} \langle\theta, x\rangle - f(x)$. A function $f : V \to \mathbb{R} \cup \{+\infty\}$ is said to be *proper* if there exists $x \in V$ such that $f(x)$ is finite. If $f$ is a proper lower semi-continuous convex function then $f^*$ is also proper lower semi-continuous convex and $f^{**} = f$.

**Coin Betting.** We consider a gambler making repeated bets on the outcomes of adversarial coin flips. The gambler starts with an initial endowment $\epsilon > 0$. In each round $t$, he bets on the outcome of a coin flip $g_t \in \{-1, 1\}$, where $+1$ denotes heads and $-1$ denotes tails. We do not make any assumption on how $g_t$ is generated, that is, it can be chosen by an adversary.

The gambler can bet any amount on either heads or tails. However, he is not allowed to borrow any additional money. If he loses, he loses the betted amount; if he wins, he gets the betted amount back and, in addition to that, he gets the same amount as a reward. We encode the gambler's bet in round $t$ by a single number $w_t$. The sign of $w_t$ encodes whether he is betting on heads or tails. The absolute value encodes the betted amount. We define $\text{Wealth}_t$ as the gambler's wealth at the end of round $t$ and $\text{Reward}_t$ as the gambler's net reward (the difference of wealth and initial endowment), that is

$$\text{Wealth}_t = \epsilon + \sum_{i=1}^{t} w_i g_i \qquad \text{and} \qquad \text{Reward}_t = \text{Wealth}_t - \epsilon . \tag{1}$$

In the following, we will also refer to a bet with $\beta_t$, where $\beta_t$ is such that

$$w_t = \beta_t \text{Wealth}_{t-1} . \tag{2}$$

The absolute value of $\beta_t$ is the *fraction* of the current wealth to bet, and sign of $\beta_t$ encodes whether he is betting on heads or tails. The constraint that the gambler cannot borrow money implies that $\beta_t \in [-1, 1]$. We also generalize the problem slightly by allowing the outcome of the coin flip $g_t$ to be any real number in the interval $[-1, 1]$; wealth and reward in (1) remain exactly the same.

# 3 Warm-Up: From Betting to One-Dimensional Online Linear Optimization

In this section, we sketch how to reduce one-dimensional OLO to betting on a coin. The reasoning for generic Hilbert spaces (Section 5) and for LEA (Section 6) will be similar. We will show that the betting view provides a natural way for the analysis and design of online learning algorithms, where the only design choice is the potential function of the betting algorithm (Section 4). A specific example of coin betting potential and the resulting algorithms are in Section 7.

As a warm-up, let us consider an algorithm for OLO over one-dimensional Hilbert space $\mathbb{R}$. Let $\{w_t\}_{t=1}^\infty$ be its sequence of predictions on a sequence of rewards $\{g_t\}_{t=1}^\infty$, $g_t \in [-1,1]$. The total reward of the algorithm after $t$ rounds is $\text{Reward}_t = \sum_{i=1}^t g_i w_i$. Also, even if in OLO there is no concept of "wealth", define the wealth of the OLO algorithm as $\text{Wealth}_t = \epsilon + \text{Reward}_t$, as in (1).

We now restrict our attention to algorithms whose predictions $w_t$ are of the form of a bet, that is $w_t = \beta_t \text{Wealth}_{t-1}$, where $\beta_t \in [-1,1]$. We will see that the restriction on $\beta_t$ does not prevent us from obtaining parameter-free algorithms with optimal bounds.

Given the above, it is immediate to see that any coin betting algorithm that, on a sequence of coin flips $\{g_t\}_{t=1}^\infty$, $g_t \in [-1,1]$, bets the amounts $w_t$ can be used as an OLO algorithm in a one-dimensional Hilbert space $\mathbb{R}$. But, what would be the regret of such OLO algorithms?

Assume that the betting algorithm at hand guarantees that its wealth is at least $F(\sum_{t=1}^T g_t)$ starting from an endowment $\epsilon$, for a given potential function $F$, then

$$\text{Reward}_T = \sum_{t=1}^T g_t w_t = \text{Wealth}_T - \epsilon \geq F\left(\sum_{t=1}^T g_t\right) - \epsilon . \tag{3}$$

Intuitively, if the reward is big we can expect the regret to be small. Indeed, the following lemma converts the lower bound on the reward to an upper bound on the regret.

**Lemma 1** (Reward-Regret relationship [22]). *Let $V, V^*$ be a pair of dual vector spaces. Let $F : V \to \mathbb{R} \cup \{+\infty\}$ be a proper convex lower semi-continuous function and let $F^* : V^* \to \mathbb{R} \cup \{+\infty\}$ be its Fenchel conjugate. Let $w_1, w_2, \ldots, w_T \in V$ and $g_1, g_2, \ldots, g_T \in V^*$. Let $\epsilon \in \mathbb{R}$. Then,*

$$\underbrace{\sum_{t=1}^T \langle g_t, w_t \rangle}_{\text{Reward}_T} \geq F\left(\sum_{t=1}^T g_t\right) - \epsilon \qquad \textit{if and only if} \qquad \forall u \in V^*, \quad \underbrace{\sum_{t=1}^T \langle g_t, u - w_t \rangle}_{\text{Regret}_T(u)} \leq F^*(u) + \epsilon .$$

Applying the lemma, we get a regret upper bound: $\text{Regret}_T(u) \leq F^*(u) + \epsilon$ for all $u \in \mathcal{H}$.

To summarize, if we have a betting algorithm that guarantees a minimum wealth of $F(\sum_{t=1}^T g_t)$, it can be used to design and analyze a one-dimensional OLO algorithm. The faster the growth of the wealth, the smaller the regret will be. Moreover, the lemma also shows that trying to design an algorithm that is adaptive to $u$ is *equivalent* to designing an algorithm that is adaptive to $\sum_{t=1}^T g_t$. Also, most importantly, *methods that guarantee optimal wealth for the betting scenario are already known*, see, e.g., [4, Chapter 9]. We can just re-use them to get optimal online algorithms!

# 4 Designing a Betting Algorithm: Coin Betting Potentials

For sequential betting on i.i.d. coin flips, an optimal strategy has been proposed by Kelly [14]. The strategy assumes that the coin flips $\{g_t\}_{t=1}^\infty$, $g_t \in \{+1, -1\}$, are generated i.i.d. with known probability of heads. If $p \in [0,1]$ is the probability of heads, the Kelly bet is to bet $\beta_t = 2p - 1$ at each round. He showed that, in the long run, this strategy will provide more wealth than betting any other fixed fraction of the current wealth [14].

For adversarial coins, Kelly betting does not make sense. With perfect knowledge of the future, the gambler could always bet everything on the right outcome. Hence, after $T$ rounds from an initial endowment $\epsilon$, the maximum wealth he would get is $\epsilon 2^T$. Instead, assume he bets the same fraction $\beta$ of its wealth at each round. Let $\text{Wealth}_t(\beta)$ the wealth of such strategy after $t$ rounds. As observed in [21], the optimal fixed fraction to bet is $\beta^* = (\sum_{t=1}^T g_t)/T$ and it gives the wealth

$$\text{Wealth}_T(\beta^*) = \epsilon \exp\left(T \cdot D\left(\frac{1}{2} + \frac{\sum_{t=1}^T g_t}{2T} \middle\| \frac{1}{2}\right)\right) \geq \epsilon \exp\left(\frac{(\sum_{t=1}^T g_t)^2}{2T}\right) , \tag{4}$$

where the inequality follows from Pinsker's inequality [9, Lemma 11.6.1].

However, even without knowledge of the future, it is possible to go very close to the wealth in (4). This problem was studied by Krichevsky and Trofimov [16], who proposed that after seeing the coin flips $g_1, g_2, \ldots, g_{t-1}$ the empirical estimate $k_t = \frac{1/2 + \sum_{i=1}^{t-1} \mathbf{1}[g_i = +1]}{t}$ should be used instead of $p$. Their estimate is commonly called *KT estimator*.[1] The KT estimator results in the betting

$$\beta_t = 2k_t - 1 = \frac{\sum_{i=1}^{t-1} g_i}{t} \tag{5}$$

which we call *adaptive Kelly betting based on the KT estimator*. It looks like an online and slightly biased version of the oracle choice of $\beta^*$. This strategy guarantees[2]

$$\text{Wealth}_T \geq \frac{\text{Wealth}_T(\beta^*)}{2\sqrt{T}} = \frac{\epsilon}{2\sqrt{T}} \exp\left(T \cdot \text{D}\left(\frac{1}{2} + \frac{\sum_{t=1}^{T} g_t}{2T} \,\middle\|\, \frac{1}{2}\right)\right) \ .$$

This guarantee is optimal up to constant factors [4] and mirrors the guarantee of the Kelly bet.

Here, we propose a new set of definitions that allows to generalize the strategy of adaptive Kelly betting based on the KT estimator. For these strategies it will be possible to prove that, for any $g_1, g_2, \ldots, g_t \in [-1, 1]$,

$$\text{Wealth}_t \geq F_t\left(\sum_{i=1}^{t} g_i\right) \ , \tag{6}$$

where $F_t(x)$ is a certain function. We call such functions *potentials*. The betting strategy will be determined uniquely by the potential (see (c) in the Definition 2), and we restrict our attention to potentials for which (6) holds. These constraints are specified in the definition below.

**Definition 2** (Coin Betting Potential). *Let $\epsilon > 0$. Let $\{F_t\}_{t=0}^{\infty}$ be a sequence of functions $F_t : (-a_t, a_t) \to \mathbb{R}_+$ where $a_t > t$. The sequence $\{F_t\}_{t=0}^{\infty}$ is called a **sequence of coin betting potentials for initial endowment** $\epsilon$, if it satisfies the following three conditions:*

*(a) $F_0(0) = \epsilon$.*

*(b) For every $t \geq 0$, $F_t(x)$ is even, logarithmically convex, strictly increasing on $[0, a_t)$, and $\lim_{x \to a_t} F_t(x) = +\infty$.*

*(c) For every $t \geq 1$, every $x \in [-(t-1), (t-1)]$ and every $g \in [-1, 1]$, $(1 + g\beta_t) F_{t-1}(x) \geq F_t(x + g)$, where*

$$\beta_t = \frac{F_t(x+1) - F_t(x-1)}{F_t(x+1) + F_t(x-1)} \ . \tag{7}$$

*The sequence $\{F_t\}_{t=0}^{\infty}$ is called a **sequence of excellent coin betting potentials for initial endowment** $\epsilon$ if it satisfies conditions (a)–(c) and the condition (d) below.*

*(d) For every $t \geq 0$, $F_t$ is twice-differentiable and satisfies $x \cdot F_t''(x) \geq F_t'(x)$ for every $x \in [0, a_t)$.*

Let's give some intuition on this definition. First, let's show by induction on $t$ that (b) and (c) of the definition together with (2) give a betting strategy that satisfies (6). The base case $t = 0$ is trivial. At time $t \geq 1$, bet $w_t = \beta_t \text{Wealth}_{t-1}$ where $\beta_t$ is defined in (7), then

$$\text{Wealth}_t = \text{Wealth}_{t-1} + w_t g_t = (1 + g_t \beta_t) \text{Wealth}_{t-1}$$

$$\geq (1 + g_t \beta_t) F_{t-1}\left(\sum_{i=1}^{t-1} g_i\right) \geq F_t\left(\sum_{i=1}^{t-1} g_i + g_t\right) = F_t\left(\sum_{i=1}^{t} g_i\right) \ .$$

The formula for the potential-based strategy (7) might seem strange. However, it is derived—see Theorem 8 in Appendix B—by minimizing the worst-case value of the right-hand side of the inequality used w.r.t. to $g_t$ in the induction proof above: $F_{t-1}(x) \geq \frac{F_t(x+g_t)}{1 + g_t \beta_t}$.

The last point, (d), is a technical condition that allows us to seamlessly reduce OLO over a Hilbert space to the one-dimensional problem, characterizing the worst case direction for the reward vectors.

Regarding the design of coin betting potentials, we expect any potential that approximates the best possible wealth in (4) to be a good candidate. In fact, $F_t(x) = \epsilon \exp\left(x^2/(2t)\right)/\sqrt{t}$, essentially the potential used in the parameter-free algorithms in [22, 24] for OLO and in [6, 18, 19] for LEA, approximates (4) and it is an excellent coin betting potential—see Theorem 9 in Appendix B. Hence, our framework provides intuition to previous constructions and in Section 7 we show new examples of coin betting potentials.

In the next two sections, we presents the reductions to effortlessly solve *both* the generic OLO case and LEA with a betting potential.

## 5  From Coin Betting to OLO over Hilbert Space

In this section, generalizing the one-dimensional construction in Section 3, we show how to use a sequence of excellent coin betting potentials $\{F_t\}_{t=0}^{\infty}$ to construct an algorithm for OLO over a Hilbert space and how to prove a regret bound for it.

We define reward and wealth analogously to the one-dimensional case: $\text{Reward}_t = \sum_{i=1}^{t}\langle g_i, w_i\rangle$ and $\text{Wealth}_t = \epsilon + \text{Reward}_t$. Given a sequence of coin betting potentials $\{F_t\}_{t=0}^{\infty}$, using (7) we define the fraction

$$\beta_t = \frac{F_t\left(\left\|\sum_{i=1}^{t-1} g_i\right\|+1\right) - F_t\left(\left\|\sum_{i=1}^{t-1} g_i\right\|-1\right)}{F_t\left(\left\|\sum_{i=1}^{t-1} g_i\right\|+1\right) + F_t\left(\left\|\sum_{i=1}^{t-1} g_i\right\|-1\right)} \ . \tag{8}$$

The prediction of the OLO algorithm is defined similarly to the one-dimensional case, but now we also need a direction in the Hilbert space:

$$w_t = \beta_t\,\text{Wealth}_{t-1}\,\frac{\sum_{i=1}^{t-1} g_i}{\left\|\sum_{i=1}^{t-1} g_i\right\|} = \beta_t\,\frac{\sum_{i=1}^{t-1} g_i}{\left\|\sum_{i=1}^{t-1} g_i\right\|}\left(\epsilon + \sum_{i=1}^{t-1}\langle g_i, w_i\rangle\right) \ . \tag{9}$$

If $\sum_{i=1}^{t-1} g_i$ is the zero vector, we define $w_t$ to be the zero vector as well. For this prediction strategy we can prove the following regret guarantee, proved in Appendix C. The proof reduces the general Hilbert case to the 1-d case, thanks to (d) in Definition 2, then it follows the reasoning of Section 3.

**Theorem 3** (Regret Bound for OLO in Hilbert Spaces). *Let $\{F_t\}_{t=0}^{\infty}$ be a sequence of excellent coin betting potentials. Let $\{g_t\}_{t=1}^{\infty}$ be any sequence of reward vectors in a Hilbert space $\mathcal{H}$ such that $\|g_t\| \leq 1$ for all $t$. Then, the algorithm that makes prediction $w_t$ defined by (9) and (8) satisfies*

$$\forall T \geq 0 \quad \forall u \in \mathcal{H} \qquad \text{Regret}_T(u) \leq F_T^*\left(\|u\|\right) + \epsilon \ .$$

## 6  From Coin Betting to Learning with Expert Advice

In this section, we show how to use the algorithm for OLO over one-dimensional Hilbert space $\mathbb{R}$ from Section 3—which is itself based on a coin betting strategy—to construct an algorithm for LEA.

Let $N \geq 2$ be the number of experts and $\Delta_N$ be the $N$-dimensional probability simplex. Let $\pi = (\pi_1, \pi_2, \ldots, \pi_N) \in \Delta_N$ be any *prior* distribution. Let $A$ be an algorithm for OLO over the one-dimensional Hilbert space $\mathbb{R}$, based on a sequence of the coin betting potentials $\{F_t\}_{t=0}^{\infty}$ with initial endowment[3] 1. We instantiate $N$ copies of $A$.

Consider any round $t$. Let $w_{t,i} \in \mathbb{R}$ be the prediction of the $i$-th copy of $A$. The LEA algorithm computes $\widehat{p}_t = (\widehat{p}_{t,1}, \widehat{p}_{t,2}, \ldots, \widehat{p}_{t,N}) \in \mathbb{R}_{0,+}^N$ as

$$\widehat{p}_{t,i} = \pi_i \cdot [w_{t,i}]_+, \tag{10}$$

where $[x]_+ = \max\{0, x\}$ is the positive part of $x$. Then, the LEA algorithm predicts $p_t = (p_{t,1}, p_{t,2}, \ldots, p_{t,N}) \in \Delta^N$ as

$$p_t = \frac{\widehat{p}_t}{\|\widehat{p}_t\|_1} \ . \tag{11}$$

If $\|\widehat{p}_t\|_1 = 0$, the algorithm predicts the prior $\pi$. Then, the algorithm receives the reward vector $g_t = (g_{t,1}, g_{t,2}, \ldots, g_{t,N}) \in [0,1]^N$. Finally, it feeds the reward to each copy of $A$. The reward for

the $i$-th copy of $A$ is $\widetilde{g}_{t,i} \in [-1,1]$ defined as

$$\widetilde{g}_{t,i} = \begin{cases} g_{t,i} - \langle g_t, p_t \rangle & \text{if } w_{t,i} > 0 \text{ ,} \\ [g_{t,i} - \langle g_t, p_t \rangle]_+ & \text{if } w_{t,i} \leq 0 \text{ .} \end{cases} \qquad (12)$$

The construction above defines a LEA algorithm defined by the predictions $p_t$, based on the algorithm $A$. We can prove the following regret bound for it.

**Theorem 4** (Regret Bound for Experts). *Let $A$ be an algorithm for OLO over the one-dimensional Hilbert space $\mathbb{R}$, based on the coin betting potentials $\{F_t\}_{t=0}^{\infty}$ for an initial endowment of $1$. Let $f_t^{-1}$ be the inverse of $f_t(x) = \ln(F_t(x))$ restricted to $[0, \infty)$. Then, the regret of the LEA algorithm with prior $\pi \in \Delta_N$ that predicts at each round with $p_t$ in (11) satisfies*

$$\forall T \geq 0 \quad \forall u \in \Delta_N \qquad \operatorname{Regret}_T(u) \leq f_T^{-1}\left(\mathrm{D}\left(u \| \pi\right)\right) \text{ .}$$

The proof, in Appendix D, is based on the fact that (10)–(12) guarantee that $\sum_{i=1}^{N} \pi_i \widetilde{g}_{t,i} w_{t,i} \leq 0$ and on a variation of the change of measure lemma used in the PAC-Bayes literature, e.g. [20].

# 7 Applications of the Krichevsky-Trofimov Estimator to OLO and LEA

In the previous sections, we have shown that a coin betting potential with a guaranteed rapid growth of the wealth will give good regret guarantees for OLO and LEA. Here, we show that the KT estimator has associated an excellent coin betting potential, which we call *KT potential*. Then, the optimal wealth guarantee of the KT potentials will translate to optimal parameter-free regret bounds.

The sequence of excellent coin betting potentials for an initial endowment $\epsilon$ corresponding to the adaptive Kelly betting strategy $\beta_t$ defined by (5) based on the KT estimator are

$$F_t(x) = \epsilon \frac{2^t \cdot \Gamma\left(\frac{t+1}{2} + \frac{x}{2}\right) \cdot \Gamma\left(\frac{t+1}{2} - \frac{x}{2}\right)}{\pi \cdot t!} \qquad t \geq 0, \quad x \in (-t-1, t+1), \qquad (13)$$

where $\Gamma(x) = \int_0^{\infty} t^{x-1} e^{-t} dt$ is Euler's gamma function—see Theorem 13 in Appendix E. This potential was used to prove regret bounds for online prediction with the logarithmic loss [16][4, Chapter 9.7]. Theorem 13 also shows that the KT betting strategy $\beta_t$ as defined by (5) satisfies (7).

This potential has the nice property that is satisfies the inequality in (c) of Definition 2 with equality when $g_t \in \{-1, 1\}$, i.e. $F_t(x + g_t) = (1 + g_t \beta_t) F_{t-1}(x)$.

We also generalize the KT potentials to $\delta$-*shifted KT potentials*, where $\delta \geq 0$, defined as

$$F_t(x) = \frac{2^t \cdot \Gamma(\delta+1) \cdot \Gamma\left(\frac{t+\delta+1}{2} + \frac{x}{2}\right) \cdot \Gamma\left(\frac{t+\delta+1}{2} - \frac{x}{2}\right)}{\Gamma\left(\frac{\delta+1}{2}\right)^2 \cdot \Gamma(t+\delta+1)} \text{ .}$$

The reason for its name is that, up to a multiplicative constant, $F_t$ is equal to the KT potential shifted in time by $\delta$. Theorem 13 also proves that the $\delta$-shifted KT potentials are excellent coin betting potentials with initial endowment $1$, and the corresponding betting fraction is $\beta_t = \frac{\sum_{j=1}^{t-1} g_j}{\delta+t}$.

## 7.1 OLO in Hilbert Space

We apply the KT potential for the construction of an OLO algorithm over a Hilbert space $\mathcal{H}$. We will use (9), and we just need to calculate $\beta_t$. According to Theorem 13 in Appendix E, the formula for $\beta_t$ simplifies to $\beta_t = \frac{\|\sum_{i=1}^{t-1} g_i\|}{t}$ so that $w_t = \frac{1}{t}\left(\epsilon + \sum_{i=1}^{t-1} \langle g_i, w_i \rangle\right) \sum_{i=1}^{t-1} g_i$.

The resulting algorithm is stated as Algorithm 1. We derive a regret bound for it as a very simple corollary of Theorem 3 to the KT potential (13). The only technical part of the proof, in Appendix F, is an upper bound on $F_t^*$ since it cannot be expressed as an elementary function.

**Corollary 5** (Regret Bound for Algorithm 1). *Let $\epsilon > 0$. Let $\{g_t\}_{t=1}^{\infty}$ be any sequence of reward vectors in a Hilbert space $\mathcal{H}$ such that $\|g_t\| \leq 1$. Then Algorithm 1 satisfies*

$$\forall T \geq 0 \quad \forall u \in \mathcal{H} \qquad \operatorname{Regret}_T(u) \leq \|u\| \sqrt{T \ln\left(1 + \frac{24 T^2 \|u\|^2}{\epsilon^2}\right)} + \epsilon\left(1 - \frac{1}{e\sqrt{\pi T}}\right) \text{ .}$$

---

**Algorithm 1** Algorithm for OLO over Hilbert space $\mathcal{H}$ based on KT potential

---
**Require:** Initial endowment $\epsilon > 0$
  1: **for** $t = 1, 2, \ldots$ **do**
  2:     Predict with $w_t \leftarrow \frac{1}{t}\left(\epsilon + \sum_{i=1}^{t-1}\langle g_i, w_i\rangle\right)\sum_{i=1}^{t-1} g_i$
  3:     Receive reward vector $g_t \in \mathcal{H}$ such that $\|g_t\| \leq 1$
  4: **end for**

---

---

**Algorithm 2** Algorithm for Learning with Expert Advice based on $\delta$-shifted KT potential

---
**Require:** Number of experts $N$, prior distribution $\pi \in \Delta_N$, number of rounds $T$
  1: **for** $t = 1, 2, \ldots, T$ **do**
  2:     For each $i \in [N]$, set $w_{t,i} \leftarrow \frac{\sum_{j=1}^{t-1}\widetilde{g}_{j,i}}{t+T/2}\left(1 + \sum_{j=1}^{t-1}\widetilde{g}_{j,i}w_{j,i}\right)$
  3:     For each $i \in [N]$, set $\widehat{p}_{t,i} \leftarrow \pi_i[w_{t,i}]_+$
  4:     Predict with $p_t \leftarrow \begin{cases} \widehat{p}_t/\|\widehat{p}_t\|_1 & \text{if } \|\widehat{p}_t\|_1 > 0 \\ \pi & \text{if } \|\widehat{p}_t\|_1 = 0 \end{cases}$
  5:     Receive reward vector $g_t \in [0,1]^N$
  6:     For each $i \in [N]$, set $\widetilde{g}_{t,i} \leftarrow \begin{cases} g_{t,i} - \langle g_t, p_t\rangle & \text{if } w_{t,i} > 0 \\ [g_{t,i} - \langle g_t, p_t\rangle]_+ & \text{if } w_{t,i} \leq 0 \end{cases}$
  7: **end for**

---

It is worth noting the elegance and extreme simplicity of Algorithm 1 and contrast it with the algorithms in [26, 22–24]. Also, the regret bound is optimal [26, 23]. The parameter $\epsilon$ can be safely set to any constant, e.g. 1. Its role is equivalent to the initial guess used in doubling tricks [25].

## 7.2   Learning with Expert Advice

We will now construct an algorithm for LEA based on the $\delta$-shifted KT potential. We set $\delta$ to $T/2$, requiring the algorithm to know the number of rounds $T$ in advance; we will fix this later with the standard doubling trick.

To use the construction in Section 6, we need an OLO algorithm for the 1-d Hilbert space $\mathbb{R}$. Using the $\delta$-shifted KT potentials, the algorithm predicts for any sequence $\{\widetilde{g}_t\}_{t=1}^{\infty}$ of reward

$$w_t = \beta_t \text{Wealth}_{t-1} = \beta_t\left(1 + \sum_{j=1}^{t-1}\widetilde{g}_j w_j\right) = \frac{\sum_{i=1}^{t-1}\widetilde{g}_i}{T/2+t}\left(1 + \sum_{j=1}^{t-1}\widetilde{g}_j w_j\right).$$

Then, following the construction in Section 6, we arrive at the final algorithm, Algorithm 2. We can derive a regret bound for Algorithm 2 by applying Theorem 4 to the $\delta$-shifted KT potential.

**Corollary 6** (Regret Bound for Algorithm 2). *Let $N \geq 2$ and $T \geq 0$ be integers. Let $\pi \in \Delta_N$ be a prior. Then Algorithm 2 with input $N, \pi, T$ for any rewards vectors $g_1, g_2, \ldots, g_T \in [0,1]^N$ satisfies*

$$\forall u \in \Delta_N \qquad \text{Regret}_T(u) \leq \sqrt{3T(3 + \text{D}(u\|\pi))}.$$

Hence, the Algorithm 2 has *both* the best known guarantee on worst-case regret and per-round time complexity, see Table 1. Also, it has the advantage of being very simple.

The proof of the corollary is in the Appendix F. The only technical part of the proof is an upper bound on $f_t^{-1}(x)$, which we conveniently do by lower bounding $F_t(x)$.

The reason for using the shifted potential comes from the analysis of $f_t^{-1}(x)$. The unshifted algorithm would have a $O(\sqrt{T(\log T + \text{D}(u\|\pi))})$ regret bound; the shifting improves the bound to $O(\sqrt{T(1 + \text{D}(u\|\pi))})$. By changing $T/2$ in Algorithm 2 to another constant fraction of $T$, it is possible to trade-off between the two constants 3 present in the square root in the regret upper bound.

The requirement of knowing the number of rounds $T$ in advance can be lifted by the standard doubling trick [25, Section 2.3.1], obtaining an anytime guarantee with a bigger leading constant,

$$\forall T \geq 0 \quad \forall u \in \Delta_N \qquad \text{Regret}_T(u) \leq \frac{\sqrt{2}}{\sqrt{2}-1}\sqrt{3T(3 + \text{D}(u\|\pi))}.$$

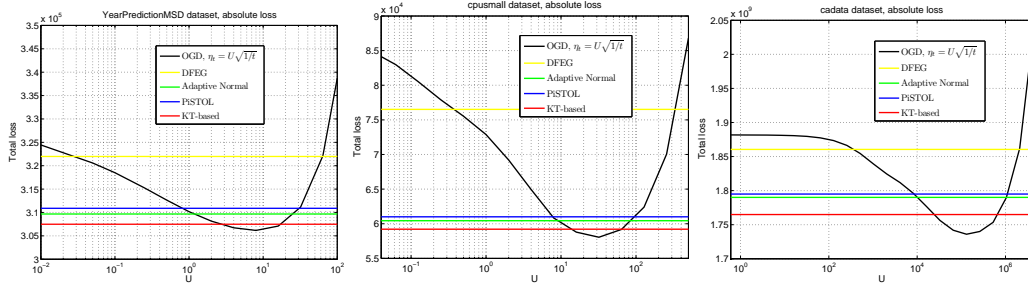

Figure 1: Total loss versus learning rate parameter of OGD (in log scale), compared with parameter-free algorithms DFEG [23], Adaptive Normal [22], PiSTOL [24] and the KT-based Algorithm 1.

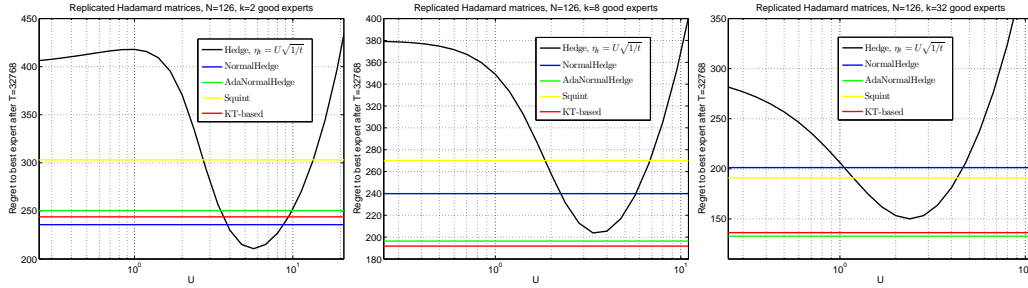

Figure 2: Regrets to the best expert after $T = 32768$ rounds, versus learning rate parameter of Hedge (in log scale). The "good" experts are $\epsilon = 0.025$ better than the others. The competitor algorithms are Normal-Hedge [6], AdaNormalHedge [19], Squint [15], and the KT-based Algorithm 2. $\pi_i = 1/N$ for all algorithms.

## 8 Discussion of the Results

We have presented a new interpretation of parameter-free algorithms as coin betting algorithms. This interpretation, far from being just a mathematical gimmick, reveals the *common* hidden structure of previous parameter-free algorithms for both OLO and LEA and also allows the design of new algorithms. For example, we show that the characteristic of parameter-freeness is just a consequence of having an algorithm that guarantees the maximum reward possible. The reductions in Sections 5 and 6 are also novel and they are in a certain sense optimal. In fact, the obtained Algorithms 1 and 2 achieve the optimal worst case upper bounds on the regret, see [26, 23] and [4] respectively.

We have also run an empirical evaluation to show that the theoretical difference between classic online learning algorithms and parameter-free ones is real and not just theoretical. In Figure 1, we have used three regression datasets[4], and solved the OCO problem through OLO. In all the three cases, we have used the absolute loss and normalized the input vectors to have L2 norm equal to 1. From the empirical results, it is clear that the optimal learning rate is completely data-dependent, yet *parameter-free algorithms have performance very close to the unknown optimal tuning of the learning rate*. Moreover, the KT-based Algorithm 1 seems to dominate all the other similar algorithms.

For LEA, we have used the synthetic setting in [6]. The dataset is composed of Hadamard matrices of size 64, where the row with constant values is removed, the rows are duplicated to 126 inverting their signs, 0.025 is subtracted to $k$ rows, and the matrix is replicated in order to generate $T = 32768$ samples. For more details, see [6]. Here, the KT-based algorithm is the one in Algorithm 2, where the term $T/2$ is removed, so that the final regret bound has an additional $\ln T$ term. Again, we see that the parameter-free algorithms have a performance close or *even better* than Hedge with an oracle tuning of the learning rate, with no clear winners among the parameter-free algorithms.

Notice that since the adaptive Kelly strategy based on KT estimator is very close to optimal, the only possible improvement is to have a data-dependent bound, for example like the ones in [24, 15, 19]. In future work, we will extend our definitions and reductions to the data-dependent case.

**Acknowledgments.** The authors thank Jacob Abernethy, Nicolò Cesa-Bianchi, Satyen Kale, Chansoo Lee, Giuseppe Molteni, and Manfred Warmuth for useful discussions on this work.

## Footnotes

[1]These algorithms require to solve a numerical problem at each step. The number $K$ is the number of steps needed to reach the required precision. Neither the precision nor $K$ are calculated in these papers.

[2]The proof in [15] can be modified to prove a KL bound, see http://blog.wouterkoolen.info.

[3]A variant of the algorithm in [11] can be implemented with the stated time complexity [10].

[1] Compared to the maximum likelihood estimate $\frac{\sum_{i=1}^{t-1} \mathbf{1}[g_i = +1]}{t-1}$, KT estimator shrinks slightly towards $1/2$.

[2] See Appendix A for a proof. For lack of space, all the appendices are in the supplementary material.

[3]Any initial endowment $\epsilon > 0$ can be rescaled to 1. Instead of $F_t(x)$ we would use $F_t(x)/\epsilon$. The $w_t$ would become $w_t/\epsilon$, but $p_t$ is invariant to scaling of $w_t$. Hence, the LEA algorithm is the same regardless of $\epsilon$.

[4]Datasets available at `https://www.csie.ntu.edu.tw/~cjlin/libsvmtools/datasets/`.

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
