[Supplementary Material]

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

## A  From Log Loss to Wealth

Guarantees for betting or sequential investement algorithm are often expressed as upper bounds on the regret with respect to the log loss. Here, for the sake of completeness, we show how to convert such a guarantee to a lower bound on the wealth of the corresponding betting algorithm.

We consider the problem of predicting a binary outcome. The algorithm predicts at each round probability $p_t \in [0,1]$. The adversary generates a sequences of outcomes $x_t \in \{0,1\}$ and the algorithm's loss is

$$\ell(p_t, x_t) = -x_t \ln p_t - (1 - x_t)\ln(1 - p_t) .$$

We define the regret with respect to a fixed probability vector $\beta$ as

$$\mathrm{Regret}_T^{\mathrm{logloss}} = \sum_{t=1}^{T} \ell(p_t, x_t) - \min_{\beta \in [0,1]} \sum_{t=1}^{T} \ell(\beta, x_t) .$$

**Lemma 7.** *Assume that an algorithm that predicts $p_t$ guarantees $\mathrm{Regret}_T^{\mathrm{logloss}} \le R_T$. Then, the coin betting strategy with endowement $\epsilon$ and $\beta_t = 2p_t - 1$ guarantees*

$$\mathrm{Wealth}_T \ge \epsilon \exp\left( T \cdot \mathrm{D}\left( \frac{1}{2} + \frac{\sum_{t=1}^{T} g_t}{2T} \middle\| \frac{1}{2} \right) - R_T \right)$$

*against any sequence of outcomes $g_t \in [-1, +1]$.*

*Proof.* Define $x_t = \frac{1+g_t}{2}$. We have

$$
\begin{aligned}
\ln \mathrm{Wealth}_T &= \ln(\mathrm{Wealth}_{t-1} + w_t g_t) \\
&= \ln(\mathrm{Wealth}_{t-1}(1 + g_t \beta_t)) \\
&= \ln \epsilon \prod_{t=1}^{T}(1 + g_t \beta_t) \\
&= \ln \epsilon + \sum_{t=1}^{T} \ln(1 + g_t \beta_t) \\
&\ge \ln \epsilon + \sum_{t=1}^{T} \left( \frac{1 + g_t}{2} \right) \ln\left(1 + \beta_t\right) + \left( \frac{1 - g_t}{2} \right) \ln\left(1 - \beta_t\right) \\
&= \ln \epsilon + \sum_{t=1}^{T} \left( \frac{1 + g_t}{2} \right) \ln\left(2 p_t\right) + \left( \frac{1 - g_t}{2} \right) \ln\left(2(1 - p_t)\right) \\
&= \ln \epsilon + T \ln(2) + \sum_{t=1}^{T} \left( \frac{1 + g_t}{2} \right) \ln(p_t) + \left( \frac{1 - g_t}{2} \right) \ln(1 - p_t) \\
&= \ln \epsilon + T \ln(2) - \sum_{t=1}^{T} \ell(p_t, x_t) \\
&= \ln \epsilon + T \ln(2) - \mathrm{Regret}_T^{\mathrm{logloss}} - \min_{\beta \in [0,1]} \sum_{t=1}^{T} \ell(\beta, x_t) \\
&\ge \ln \epsilon + T \ln(2) - R_T - \min_{\beta \in [0,1]} \sum_{t=1}^{T} \ell(\beta, x_t) ,
\end{aligned}
$$

where the first inequality is due to the concavity of $\ln$ and the second one is due to the assumption of the regret.

It is easy to see that the $\beta^* = \arg\min_{\beta \in [0,1]} \sum_{t=1}^{T} \ell(\beta, x_t) = \frac{\sum_{t=1}^{T} x_t}{T}$. Hence, we have

$$\min_{\beta \in [0,1]} \sum_{t=1}^{T} \ell(\beta, x_t) = T\left(-\beta^* \ln \beta^* - (1 - \beta^*)\ln(1 - \beta^*)\right) .$$

Also, we have that for any $\beta \in [0, 1]$

$$-\beta \ln \beta - (1 - \beta) \ln(1 - \beta) = -\mathrm{D}\left(\beta \left\| \frac{1}{2}\right.\right) + \ln 2 \ .$$

Putting all together, we have the stated lemma. □

The lower bound on the wealth of the adaptive Kelly betting based on the KT estimator is obtained simply by the stated Lemma and reminding that the log loss regret of the KT estimator is upper bounded by $\frac{1}{2} \ln T + \ln 2$.

## B  Optimal Betting Fraction

**Theorem 8** (Optimal Betting Fraction). *Let $x \in \mathbb{R}$. Let $F : [x - 1, x + 1] \to \mathbb{R}$ be a logarithmically convex function. Then,*

$$\underset{\beta \in (-1,1)}{\arg \min} \ \underset{g \in [-1,1]}{\max} \ \frac{F(x + g)}{1 + \beta g} = \frac{F(x + 1) - F(x - 1)}{F(x + 1) + F(x - 1)} \ .$$

*Moreover, $\beta^* = \frac{F(x+1) - F(x-1)}{F(x+1) + F(x-1)}$ satisfies*

$$\ln(F(x + 1)) - \ln(1 + \beta^*) = \ln(F(x - 1)) - \ln(1 - \beta^*) \ .$$

*Proof.* We define the functions $h, f : [-1, 1] \times (-1, 1) \to \mathbb{R}$ as

$$h(g, \beta) = \frac{F(x + g)}{1 + \beta g} \qquad \text{and} \qquad f(g, \beta) = \ln(h(g, \beta)) = \ln(F(x + g)) - \ln(1 + \beta g) \ .$$

Clearly, $\arg \min_{\beta \in (-1,1)} \max_{g \in [-1,1]} h(g, \beta) = \arg \min_{\beta \in (-1,1)} \max_{g \in [-1,1]} f(g, \beta)$ and we can work with $f$ instead of $h$. The function $h$ is logarithmically convex in $g$ and thus $f$ is convex in $g$. Therefore,

$$\forall \beta \in (-1, 1) \qquad \max_{g \in [-1,1]} f(g, \beta) = \max \left\{ f(+1, \beta), f(-1, \beta) \right\} \ .$$

Let $\phi(\beta) = \max \left\{ f(+1, \beta), f(-1, \beta) \right\}$. We seek to find the $\arg \min_{\beta \in (-1,1)} \phi(\beta)$. Since $f(+1, \beta)$ is decreasing in $\beta$ and $f(-1, \beta)$ is increasing in $\beta$, the minimum of $\phi(\beta)$ is at a point $\beta^*$ such that $f(+1, \beta^*) = f(-1, \beta^*)$. In other words, $\beta^*$ satisfies

$$\ln(F(x + 1)) - \ln(1 + \beta^*) = \ln(F(x - 1)) - \ln(1 - \beta^*) \ .$$

The only solution of this equation is

$$\beta^* = \frac{F(x + 1) - F(x - 1)}{F(x + 1) + F(x - 1)} \ .$$

□

**Theorem 9.** *The functions $F_t(x) = \epsilon \exp(\frac{x^2}{2t} - \frac{1}{2} \sum_{i=1}^t \frac{1}{i})$ are excellent coin betting potentials.*

*Proof.* The first and second properties of Definition 2 are trivially true. For the third property, we first use Theorem 8 to have

$$\ln(1 + \beta_t g) - \ln F_t(x + g) \geq \ln(1 + \beta_t) - \ln F_t(x + 1) = \ln \frac{2}{F_t(x + 1) + F_t(x - 1)},$$

where the definition of $\beta_t$ is from (7). Hence, we have

$$\ln(1 + \beta_t g) - \ln F_t(x + g) + \ln F_{t-1}(x) \geq \ln \frac{2}{F_t(x+1) + F_t(x-1)} + \ln F_{t-1}(x)$$

$$= -\frac{x^2 + 1}{2t} + \frac{1}{2}\sum_{i=1}^{t}\frac{1}{i} - \ln \cosh \frac{x}{t} + \frac{x^2}{2(t-1)} - \frac{1}{2}\sum_{i=1}^{t-1}\frac{1}{i}$$

$$= -\frac{x^2}{2t} - \ln \cosh \frac{x}{t} + \frac{x^2}{2(t-1)}$$

$$\geq -\frac{x^2}{2t} - \frac{x^2}{2t^2} + \frac{x^2}{2(t-1)}$$

$$\geq -\frac{x^2}{2t} - \frac{x^2}{2t(t-1)} + \frac{x^2}{2(t-1)} = 0,$$

where in the second inequality we have used the elementary inequality $\ln \cosh x \leq \frac{x^2}{2}$.

The fourth property of Definition 2 is also true because $F_t(x)$ is of the form $h(x^2)$ with $h(\cdot)$ convex [22].  $\qquad \square$

## C   Proof of Lemma 11

First we state the following Lemma from [22] and reported here with our notation for completeness.

**Lemma 10** (Extremes). *Let $h : (-a, a) \to \mathbb{R}$ be an even twice-differentiable function that satisfies $x \cdot h''(x) \geq h'(x)$ for all $x \in [0, a)$. Let $c : [0, \infty) \times [0, \infty) \to \mathbb{R}$ be an arbitrary function. Then, if vectors $u, v \in \mathcal{H}$ satisfy $\|u\| + \|v\| < a$, then*

$$c(\|u\|, \|v\|) \cdot \langle u, v \rangle - h(\|u + v\|) \geq \min \{ c(\|u\|, \|v\|) \cdot \|u\| \cdot \|v\| - h(\|v\| + \|u\|),$$
$$-c(\|u\|, \|v\|) \cdot \|u\| \cdot \|v\| - h(\|u\| - \|v\|) \} . \quad (14)$$

*Proof.* If $u$ or $v$ is zero, the inequality (14) clearly holds. From now on we assume that $u, v$ are non-zero. Let $\alpha$ be the cosine of the angle of between $u$ and $v$. More formally,

$$\alpha = \frac{\langle u, v \rangle}{\|u\| \cdot \|v\|} .$$

With this notation, the left-hand side of (14) is

$$f(\alpha) = c(\|u\|, \|v\|) \cdot \alpha \|u\| \cdot \|v\| - h(\sqrt{\|u\|^2 + \|v\|^2 + 2\alpha \|u\| \cdot \|v\|}) .$$

Since $h$ is even, the inequality (14) is equivalent to

$$\forall \alpha \in [-1, 1] \qquad f(\alpha) \geq \min \{ f(+1), f(-1) \} .$$

The last inequality is clearly true if $f : [-1, 1] \to \mathbb{R}$ is concave. We now check that $f$ is indeed concave, which we prove by showing that the second derivative is non-positive. The first derivative of $f$ is

$$f'(\alpha) = c(\|u\|, \|v\|) \cdot \|u\| \cdot \|v\| - \frac{h'(\sqrt{\|u\|^2 + \|v\|^2 + 2\alpha \|u\| \cdot \|v\|}) \cdot \|u\| \cdot \|v\|}{\sqrt{\|u\|^2 + \|v\|^2 + 2\alpha \|u\| \cdot \|v\|}} .$$

The second derivative of $f$ is

$$f''(\alpha) = -\frac{\|u\|^2 \cdot \|v\|^2}{\|u\|^2 + \|v\|^2 + 2\alpha \|u\| \cdot \|v\|}$$
$$\cdot \left( h''(\sqrt{\|u\|^2 + \|v\|^2 + 2\alpha \|u\| \cdot \|v\|}) - \frac{h'(\sqrt{\|u\|^2 + \|v\|^2 + 2\alpha \|u\| \cdot \|v\|})}{\sqrt{\|u\|^2 + \|v\|^2 + 2\alpha \|u\| \cdot \|v\|}} \right) .$$

If we consider $x = \sqrt{\|u\|^2 + \|v\|^2 + 2\alpha \|u\| \cdot \|v\|}$, the assumption $x \cdot h''(x) \geq h'(x)$ implies that $f''(\alpha)$ is non-positive. This finishes the proof of the inequality (14).  $\qquad \square$

We also need the following technical Lemma whose proof relies mainly on property (d) of Definition 2.

**Lemma 11.** *Let $\{F_t\}_{t=0}^\infty$ be a sequence of excellent coin betting potentials. Let $g_1, g_2, \ldots, g_t$ be vectors in a Hilbert space $\mathcal{H}$ such that $\|g_1\|, \|g_2\|, \ldots, \|g_t\| \leq 1$. Let $\beta_t$ be defined by (8) and let $x = \sum_{i=1}^{t-1} g_i$. Then,*

$$\left(1 + \beta_t \frac{\langle g_t, x \rangle}{\|x\|}\right) F_{t-1}(\|x\|) \geq F_t(\|x + g_t\|) .$$

*Proof.* Since $F_t(x)$ is an excellent coin betting potential, it satisfies $xF_t''(x) \geq F_t'(x)$. Hence,

$$\left(1 + \beta_t \frac{\langle g_t, x \rangle}{\|x\|}\right) F_{t-1}(\|x\|) - F_t(\|x + g_t\|)$$

$$= F_{t-1}(\|x\|) + \beta_t \frac{\langle g_t, x \rangle}{\|x\|} F_{t-1}(\|x\|) - F_t(\|x + g_t\|)$$

$$\geq F_{t-1}(\|x\|) + \min_{r \in \{-1,1\}} \beta_t r \|g_t\| F_{t-1}(\|x\|) - F_t(\|x\| + r \|g_t\|)$$

$$= \min_{r \in \{-1,1\}} (1 + \beta_t r \|g_t\|) F_{t-1}(\|x\|) - F_t(\|x\| + r \|g_t\|)$$

$$\geq 0 .$$

If $x \neq 0$, the first inequality comes from Lemma 10 with $c(z, \cdot) = \frac{F_{t-1}(z+1) - F_{t-1}(z-1)}{F_{t-1}(z+1) + F_{t-1}(z-1)} F_{t-1}(z)/z$ and $h(z) = F_t(z)$, $u = g_t$, $v = x$. If $x = 0$ then, according to (8), $\beta_t = 0$ and the first inequality trivially holds. The second inequality follows from the property (c) of a coin betting potential. $\square$

*Proof of Theorem 3.* First, by induction on $t$ we show that

$$\text{Wealth}_t \geq F_t \left( \left\| \sum_{t=1}^T g_t \right\| \right) . \tag{15}$$

The base case $t = 0$ is trivial, since both sides of the inequality are equal to $\epsilon$. For $t \geq 1$, if we let $x = \sum_{i=1}^{t-1} g_i$, we have

$$\text{Wealth}_t = \langle g_t, w_t \rangle + \text{Wealth}_{t-1} = \left(1 + \beta_t \frac{\langle g_t, x \rangle}{\|x\|}\right) \text{Wealth}_{t-1}$$

$$\geq \left(1 + \beta_t \frac{\langle g_t, x \rangle}{\|x\|}\right) F_{t-1}(\|x\|) \overset{(*)}{\geq} F_t(\|x + g_t\|) = F_t \left( \left\| \sum_{i=1}^t g_i \right\| \right) .$$

The inequality marked with $(*)$ follows from Lemma 11.

This establishes (15), from which we immediately have a reward lower bound

$$\text{Reward}_T = \sum_{t=1}^T \langle g_t, w_t \rangle = \text{Wealth}_T - \epsilon \geq F_T \left( \left\| \sum_{t=1}^T g_t \right\| \right) - \epsilon . \tag{16}$$

We apply Lemma 1 to the function $F(x) = F_T(\|x\|) - \epsilon$ and we are almost done. The only remaining property we need is that if $F$ is an even function then the Fenchel conjugate of $F(\|\cdot\|)$ is $F^*(\|\cdot\|)$; see Bauschke and Combettes [3, Example 13.7]. $\square$

# D  Proof of Theorem 4

*Proof.* We first prove that $\sum_{i=1}^{N} \pi_i \widetilde{g}_{t,i} w_{t,i} \leq 0$. Indeed,

$$\sum_{i=1}^{N} \pi_i \widetilde{g}_{t,i} w_{t,i} = \sum_{i \,:\, \pi_i w_{t,i} > 0} \pi_i [w_{t,i}]_+ (g_{t,i} - \langle g_t, p_t \rangle) + \sum_{i \,:\, \pi_i w_{t,i} \leq 0} \pi_i w_{t,i} [g_{t,i} - \langle g_t, p_t \rangle]_+$$

$$= \|\widehat{p}_t\|_1 \sum_{i=1}^{N} p_{t,i} (g_{t,i} - \langle g_t, p_t \rangle) + \sum_{i \,:\, \pi_i w_{t,i} \leq 0} \pi_i w_{t,i} [g_{t,i} - \langle g_t, p_t \rangle]_+$$

$$= 0 + \sum_{i \,:\, \pi_i w_{t,i} \leq 0} \pi_i w_{t,i} [g_{t,i} - \langle g_t, p_t \rangle]_+ \leq 0 .$$

The first equality follows from definition of $g_{t,i}$. To see the second equality, consider two cases: If $\pi_i w_{t,i} \leq 0$ for all $i$ then $\|\widehat{p}_t\|_1 = 0$ and therefore both $\|\widehat{p}_t\|_1 \sum_{i=1}^{N} p_{t,i}(g_{t,i} - \langle g_t, p_t \rangle)$ and $\sum_{i \,:\, \pi_i w_{t,i} > 0} \pi_i [w_{t,i}]_+ (g_{t,i} - \langle g_t, p_t \rangle)$ are trivially zero. If $\|\widehat{p}_t\|_1 > 0$ then $\pi_i [w_{t,i}]_+ = \widehat{p}_{t,i} = \|\widehat{p}_t\|_1 p_{t,i}$ for all $i$.

From the assumption on $A$, we have, for any sequence $\{\widetilde{g}_t\}_{t=1}^{\infty}$ such that $\widetilde{g}_t \in [-1, 1]$, satisfies

$$\text{Wealth}_t = 1 + \sum_{i=1}^{t} \widetilde{g}_i w_i \geq F_t \left( \sum_{i=1}^{t} \widetilde{g}_i \right) . \tag{17}$$

Inequality $\sum_{i=1}^{N} \pi_i \widetilde{g}_{t,i} w_{t,i} \leq 0$ and (17) imply

$$\sum_{i=1}^{N} \pi_i F_T \left( \sum_{t=1}^{T} \widetilde{g}_{t,i} \right) \leq 1 + \sum_{i=1}^{N} \pi_i \sum_{t=1}^{T} \widetilde{g}_{t,i} w_{t,i} \leq 1 . \tag{18}$$

Now, let $\widetilde{G}_{T,i} = \sum_{t=1}^{T} \widetilde{g}_{t,i}$. For any competitor $u \in \Delta_N$,

$$\text{Regret}_T(u) = \sum_{t=1}^{T} \langle g_t, u - p_t \rangle = \sum_{t=1}^{T} \sum_{i=1}^{N} u_i (g_{t,i} - \langle g_t, p_t \rangle)$$

$$\leq \sum_{t=1}^{T} \sum_{i=1}^{N} u_i \widetilde{g}_{t,i} \qquad \text{(by definition of } \widetilde{g}_{t,i})$$

$$\leq \sum_{i=1}^{N} u_i \left| \widetilde{G}_{T,i} \right| \qquad \text{(since } u_i \geq 0, i = 1, \ldots, N)$$

$$= \sum_{i=1}^{N} u_i f_T^{-1} \left( \ln[F_T(\widetilde{G}_{T,i})] \right) \qquad \text{(since } F_T(x) = \exp(f_T(x)) \text{ is even)}$$

$$\leq f_T^{-1} \left( \sum_{i=1}^{N} u_i \ln \left[ F_T(\widetilde{G}_{T,i}) \right] \right) \qquad \text{(by concavity of } f_T^{-1})$$

$$= f_T^{-1} \left( \sum_{i=1}^{N} u_i \left\{ \ln \left[ \frac{u_i}{\pi_i} \right] + \ln \left[ \frac{\pi_i}{u_i} F_T(\widetilde{G}_{T,i}) \right] \right\} \right) = f_T^{-1} \left( D\left( u \| \pi \right) + \sum_{i=1}^{N} u_i \ln \left[ \frac{\pi_i}{u_i} F_T(\widetilde{G}_{T,i}) \right] \right)$$

$$\leq f_T^{-1} \left( D\left( u \| \pi \right) + \ln \left( \sum_{i=1}^{N} \pi_i F_T(\widetilde{G}_{T,i}) \right) \right) \qquad \text{(by concavity of } \ln(\cdot))$$

$$\leq f_T^{-1} \left( D\left( u \| \pi \right) \right) \qquad \text{(by (18))}. \qquad \square$$

# E  Properties of Krichevsky-Trofimov Potential

**Lemma 12** (Analytic Properties of KT potential). *Let $a > 0$. The function $F : (-a, a) \to \mathbb{R}_+$,*

$$F(x) = \Gamma(a + x)\Gamma(a - x)$$

*is even, logarithmically convex, strictly increasing on* $[0, a)$*, satisfies*

$$\lim_{x \nearrow a} F(x) = \lim_{x \searrow -a} F(x) = +\infty$$

*and*

$$\forall x \in [0, a) \qquad x \cdot F''(x) \geq F'(x) . \tag{19}$$

*Proof.* $F(x)$ is obviously even. $\Gamma(z) = \int_0^\infty t^{z-1} e^{-t} dt$ is defined for any real number $z > 0$. Hence, $F$ is defined on the interval $(-a, a)$. According to Bohr-Mollerup theorem [1, Theorem 2.1], $\Gamma(x)$ is logarithmically convex on $(0, \infty)$. Hence, $F(x)$ is also logarithmically convex, since $\ln(F(x)) = \ln(\Gamma(a+x)) + \ln(\Gamma(a-x))$ is a sum of convex functions.

It is well known that $\lim_{z \searrow 0} \Gamma(z) = +\infty$. Thus,

$$\lim_{x \nearrow a} F(x) = \lim_{x \nearrow a} \Gamma(a+x)\Gamma(a-x) = \Gamma(2a) \lim_{x \nearrow a} \Gamma(a-x) = \Gamma(2a) \lim_{z \searrow 0} \Gamma(z) = +\infty ,$$

since $\Gamma$ is continuous and not zero at $2a$. Because $F(x)$ is even, we also have $\lim_{x \searrow -a} F(x) = +\infty$.

To show that $F(x)$ is increasing and that it satisfies (19), we write $f(x) = \ln(F(x))$ as a Mclaurin series. The derivatives of $\ln(\Gamma(z))$ are the so called polygamma functions

$$\psi^{(n)}(z) = \frac{d^{n+1}}{dz^{n+1}} \ln(\Gamma(z)) \qquad \text{for } z > 0 \text{ and } n = 0, 1, 2, \ldots.$$

Polygamma functions have the well-known integral representation

$$\psi^{(n)}(z) = (-1)^{n+1} \int_0^\infty \frac{t^n e^{-zt}}{1 - e^{-t}} dt \qquad \text{for } z > 0 \text{ and } n = 1, 2, \ldots.$$

Using polygamma functions, we can write the Mclaurin series for $f(x) = \ln(F(x))$ as

$$f(x) = \ln(F(x)) = \ln(\Gamma(a+x)) + \ln(\Gamma(a-x)) = 2\ln(\Gamma(a)) + 2 \sum_{\substack{n \geq 2 \\ n \text{ even}}} \frac{\psi^{(n-1)}(a) x^n}{n!} .$$

The series converges for $x \in (-a, a)$, since for even $n \geq 2$, $\psi^{(n-1)}(a)$ is positive and can be upper bounded as

$$\begin{aligned}
\psi^{(n-1)}(a) &= \int_0^\infty \frac{t^{n-1} e^{-at}}{1 - e^{-t}} dt \\
&= \int_0^1 \frac{t^{n-1} e^{-at}}{1 - e^{-t}} dt + \int_1^\infty \frac{t^{n-1} e^{-zt}}{1 - e^{-t}} dt \\
&\leq \int_0^1 \frac{t^{n-1} e^{-at}}{t(1 - 1/e)} dt + \int_1^\infty t^{n-1} e^{-at} dt \\
&\leq \frac{1}{1 - 1/e} \int_0^\infty t^{n-2} e^{-at} dt + \int_0^\infty t^{n-1} e^{-at} dt \\
&= \frac{1}{1 - 1/e} a^{1-n} \Gamma(n-1) + a^{-n} \Gamma(n) \\
&\leq \frac{1}{1 - 1/e} a^{-n} (a+1)(n-1)! .
\end{aligned}$$

From the Mclaurin expansion we see that $f(x)$ is increasing on $[0, a)$ since all the coefficients are positive (except for zero order term).

Finally, to prove (19), note that for any $x \in (-a, a)$,

$$f(x) = c_0 + \sum_{n=2}^\infty c_n x^n$$

where $c_2, c_3, \ldots$ are non-negative coefficients. Thus

$$f'(x) = \sum_{n=2}^\infty n c_n x^{n-1} \qquad \text{and} \qquad f''(x) = \sum_{n=2}^\infty n(n-1) c_n x^{n-2} .$$

and hence $x \cdot f''(x) \geq f'(x)$ for $x \in [0, a)$. Since $F(x) = \exp(f(x))$,

$$F'(x) = f'(x) \cdot F(x) \qquad \text{and} \qquad F''(x) = \left[ f''(x) + (f'(x))^2 \right] \cdot F(x) .$$

Therefore, for $x \in [0, a)$,

$$x \cdot F''(x) = x \left[ f''(x) + (f'(x))^2 \right] F(x) \geq \left[ f'(x) + x(f'(x))^2 \right] F(x) \geq f'(x) F(x) = F'(x) .$$

This proves (19). $\qquad\qquad\qquad\qquad\qquad\qquad\qquad\qquad\qquad\qquad\qquad\qquad\qquad\qquad\qquad\qquad$ $\square$

**Theorem 13** (KT potential). *Let $\delta \geq 0$ and $\epsilon > 0$. The sequence of functions $\{F_t\}_{t=0}^{\infty}$, $F_t :$ $(-t - \delta - 1, t + \delta + 1) \to \mathbb{R}_+$ defined by*

$$F_t(x) = \epsilon \frac{2^t \cdot \Gamma(\delta + 1) \Gamma(\frac{t+\delta+1}{2} + \frac{x}{2}) \Gamma(\frac{t+\delta+1}{2} - \frac{x}{2})}{\Gamma(\frac{\delta+1}{2})^2 \Gamma(t + \delta + 1)} .$$

*is a sequence of excellent coin betting potentials for initial endowment $\epsilon$. Furthermore, for any $x \in (-t - \delta - 1, t + \delta + 1)$,*

$$\frac{F_t(x+1) - F_t(x-1)}{F_t(x+1) + F_t(x-1)} = \frac{x}{t+\delta} . \tag{20}$$

*Proof.* Property (b) and (d) of the definition follow from Lemma 12. Property (a) follows by simple substitution for $t = 0$ and $x = 0$.

Before verifying property (c), we prove (20). We use an algebraic property of the gamma function that states that $\Gamma(1 + z) = z\Gamma(z)$ for any positive $z$. Equation (20) follows from

$$\begin{aligned}
\frac{F_t(x+1) - F_t(x-1)}{F_t(x+1) + F_t(x-1)} &= \frac{\Gamma(\frac{t+\delta+2}{2} + \frac{x}{2})\Gamma(\frac{t+\delta}{2} - \frac{x}{2}) - \Gamma(\frac{t+\delta}{2} + \frac{x}{2})\Gamma(\frac{t+\delta+2}{2} - \frac{x}{2})}{\Gamma(\frac{t+\delta+2}{2} + \frac{x}{2})\Gamma(\frac{t+\delta}{2} - \frac{x}{2}) + \Gamma(\frac{t+\delta}{2} + \frac{x}{2})\Gamma(\frac{t+\delta+2}{2} - \frac{x}{2})} \\
&= \frac{(\frac{t+\delta}{2} + \frac{x}{2})\Gamma(\frac{t+\delta}{2} + \frac{x}{2})\Gamma(\frac{t+\delta}{2} - \frac{x}{2}) - (\frac{t+\delta}{2} - \frac{x}{2})\Gamma(\frac{t+\delta}{2} + \frac{x}{2})\Gamma(\frac{t+\delta}{2} - \frac{x}{2})}{(\frac{t+\delta}{2} + \frac{x}{2})\Gamma(\frac{t+\delta}{2} + \frac{x}{2})\Gamma(\frac{t+\delta}{2} - \frac{x}{2}) + (\frac{t+\delta}{2} - \frac{x}{2})\Gamma(\frac{t+\delta}{2} + \frac{x}{2})\Gamma(\frac{t+\delta}{2} - \frac{x}{2})} \\
&= \frac{(\frac{t+\delta}{2} + \frac{x}{2}) - (\frac{t+\delta}{2} - \frac{x}{2})}{(\frac{t+\delta}{2} + \frac{x}{2}) + (\frac{t+\delta}{2} - \frac{x}{2})} \\
&= \frac{x}{t+\delta} .
\end{aligned}$$

Let $\phi(g) = \frac{F_t(x+g)}{F_{t-1}(x)}$. To verify property (c) of the definition, we need to show that $\phi(g) \leq 1 + g\frac{x}{t+\delta}$ for any $x \in [-t + 1, t - 1]$ and any $g \in [-1, 1]$. We can write $\phi(g)$ as

$$\begin{aligned}
\phi(g) &= \frac{F_t(x+g)}{F_{t-1}(x)} \\
&= \frac{2\Gamma(\frac{t+\delta+1}{2} + \frac{x+g}{2})\Gamma(\frac{t+\delta+1}{2} - \frac{x+g}{2})\Gamma(t+\delta)}{\Gamma(\frac{t+\delta}{2} + \frac{x}{2})\Gamma(\frac{t+\delta}{2} - \frac{x}{2})\Gamma(t + \delta + 1)} \\
&= \frac{2}{t+\delta} \cdot \frac{\Gamma(\frac{t+\delta+1}{2} + \frac{x+g}{2})\Gamma(\frac{t+\delta+1}{2} - \frac{x+g}{2})}{\Gamma(\frac{t+\delta}{2} + \frac{x}{2})\Gamma(\frac{t+\delta}{2} - \frac{x}{2})} .
\end{aligned}$$

For $g = +1$, using the formula $\Gamma(1 + z) = z\Gamma(z)$, we have

$$\phi(+1) = \frac{2}{t+\delta} \cdot \frac{\Gamma(\frac{t+\delta}{2} + \frac{x}{2} + 1)\Gamma(\frac{t+\delta}{2} - \frac{x}{2})}{\Gamma(\frac{t+\delta}{2} + \frac{x}{2})\Gamma(\frac{t+\delta}{2} - \frac{x}{2})} = \frac{2}{t+\delta}\left( \frac{t+\delta}{2} + \frac{x}{2} \right) = 1 + \frac{x}{t+\delta} .$$

Similarly, for $g = -1$, using the formula $\Gamma(1 + z) = z\Gamma(z)$, we have

$$\phi(-1) = \frac{2}{t+\delta} \cdot \frac{\Gamma(\frac{t+\delta}{2} + \frac{x}{2})\Gamma(\frac{t+\delta}{2} - \frac{x}{2} + 1)}{\Gamma(\frac{t+\delta}{2} + \frac{x}{2})\Gamma(\frac{t+\delta}{2} - \frac{x}{2})} = \frac{2}{t+\delta}\left( \frac{t+\delta}{2} - \frac{x}{2} \right) = 1 - \frac{x}{t+\delta} .$$

We can write any $g \in [-1, 1]$ as a convex combination of $-1$ and $+1$, i.e., $g = \lambda \cdot (-1) + (1 - \lambda) \cdot (+1)$ for some $\lambda \in [0, 1]$. Since $\phi(g)$ is (logarithmically) convex,

$$
\begin{aligned}
\phi(g) &= \phi(\lambda \cdot (-1) + (1 - \lambda) \cdot (+1)) \\
&\leq \lambda \phi(-1) + (1 - \lambda) \phi(+1) \\
&= \lambda \left(1 + \frac{x}{t + \delta}\right) + (1 - \lambda) \left(1 - \frac{x}{t + \delta}\right) \\
&= 1 + g \frac{x}{t + \delta} \,.
\end{aligned}
$$
$\square$

# F   Proofs of Corollaries 5 and 6

We state some technical lemmas that will be used in the following proofs. We start with a lower bound on the Krichevsky-Trofimov (KT) potential. It is a generalization of the lower bound proved for integers in Willems et al. [29] to real numbers.

**Lemma 14** (Lower Bound on KT Potential). *If $c \geq 1$ and $a, b$ are non-negative reals such that $a + b = c$ then*

$$
\ln\left(\frac{\Gamma(a + 1/2) \cdot \Gamma(b + 1/2)}{\pi \cdot \Gamma(c + 1)}\right) \geq -\ln(e\sqrt{\pi}) - \frac{1}{2}\ln(c) + \ln\left(\left(\frac{a}{c}\right)^a \left(\frac{b}{c}\right)^b\right) \,.
$$

*Proof.* From [28][p. 263 Ex. 45], we have

$$
\frac{\Gamma(a + 1/2)\Gamma(b + 1/2)}{\Gamma(a + b + 1)} \geq \sqrt{2\pi} \frac{(a + 1/2)^a (b + 1/2)^b}{(a + b + 1)^{a + b + 1/2}} \,.
$$

It remains to show that

$$
\sqrt{2\pi} \frac{(a + 1/2)^a (b + 1/2)^b}{(a + b + 1)^{a + b + 1/2}} > \frac{\sqrt{\pi}}{e} \frac{1}{\sqrt{a + b}} \left(\frac{a}{a + b}\right)^a \left(\frac{b}{a + b}\right)^b \,,
$$

which is equivalent to

$$
\frac{(1 + \frac{1}{2a})^a (1 + \frac{1}{2b})^b}{(1 + \frac{1}{a + b})^{a + b + 1/2}} > \frac{1}{e\sqrt{2}} \,.
$$

From the inequality $1 \leq (1 + 1/x)^x < e$ valid for any $x \geq 0$, it follows that $1 \leq (1 + \frac{1}{2a})^a < \sqrt{e}$ and $1 \leq (1 + \frac{1}{2b})^b < \sqrt{e}$ and $1 \leq (1 + 1/(a + b))^{a + b} < e$. Hence,

$$
\frac{(1 + \frac{1}{2a})^a (1 + \frac{1}{2b})^b}{(1 + \frac{1}{a + b})^{a + b + 1/2}} > \frac{1}{e\sqrt{1 + \frac{1}{a + b}}} \geq \frac{1}{e\sqrt{2}} \,.
$$
$\square$

**Lemma 15.** *Let $\delta \geq 0$. Then*

$$
\frac{\Gamma(\delta + 1)}{2^\delta \Gamma(\frac{\delta + 1}{2})^2} \geq \frac{\sqrt{\delta + 1}}{\pi} \,.
$$

*Proof.* We will prove the equivalent statement that

$$
\ln \frac{\Gamma(\delta + 1)\pi}{2^\delta \Gamma(\frac{\delta + 1}{2})^2 \sqrt{\delta + 1}} \geq 0 \,.
$$

The inequality holds with equality in $\delta = 0$, so it is enough to prove that the derivative of the left-hand side is positive for $\delta > 0$. The derivative of the left-hand side is equal to

$$
\Psi(\delta + 1) - \frac{1}{2(\delta + 1)} - \ln(2) - \Psi\left(\frac{\delta + 1}{2}\right) \,,
$$

where $\Psi(x)$ is the digamma function.

We will use the upper [7] and lower bound [2] to the digamma function, which state that for any $x > 0$,

$$\Psi(x) < \ln(x) - \frac{1}{2x} - \frac{1}{12x^2} + \frac{1}{120x^4}$$

$$\Psi(x+1) > \ln\left(x + \frac{1}{2}\right) .$$

Using these bounds we have

$$\Psi(\delta + 1) - \frac{1}{2(\delta + 1)} - \ln(2) - \Psi\left(\frac{\delta + 1}{2}\right)$$

$$\geq \ln\left(\delta + \frac{1}{2}\right) - \frac{1}{2(\delta + 1)} - \ln(2) - \ln\left(\frac{\delta + 1}{2}\right) + \frac{1}{\delta + 1} + \frac{1}{3(\delta + 1)^2} - \frac{2}{15(\delta + 1)^4}$$

$$= \ln\left(1 - \frac{1}{2(\delta + 1)}\right) + \frac{1}{2(\delta + 1)} + \frac{1}{3(\delta + 1)^2} - \frac{2}{15(\delta + 1)^4}$$

$$\geq -\frac{(4\ln(2) - 2)}{4(\delta + 1)^2} + \frac{1}{3(\delta + 1)^2} - \frac{2}{15(\delta + 1)^4}$$

$$= \frac{[15(1/2 - \ln(2))) + 5](\delta + 1)^2 - 2}{15(\delta + 1)^4}$$

$$\geq \frac{[15(1/2 - \ln(2))) + 5] - 2}{15(\delta + 1)^4} \geq 0$$

where in the second inequality we used the elementary inequality $\ln(1 - x) \geq -x - (4\ln(2) - 2)x^2$ valid for $x \in [0, .5]$. $\qquad \square$

**Lemma 16** (Lower Bound on Shifted KT Potential). *Let $T \geq 1$, $\delta \geq 0$, and $x \in [-T, T]$. Then*

$$\frac{2^T \cdot \Gamma(\delta + 1)\Gamma\left(\frac{T + \delta + 1}{2} + \frac{x}{2}\right) \cdot \Gamma\left(\frac{T + \delta + 1}{2} - \frac{x}{2}\right)}{\Gamma\left(\frac{\delta + 1}{2}\right)^2 \Gamma(T + \delta + 1)} \geq \exp\left(\frac{x^2}{2(T + \delta)} + \frac{1}{2}\ln\left(\frac{1 + \delta}{T + \delta}\right) - \ln(e\sqrt{\pi})\right) .$$

*Proof.* Using Lemma 14, we have

$$\ln \frac{2^T \cdot \Gamma(\delta + 1)\Gamma\left(\frac{T + \delta + 1}{2} + \frac{x}{2}\right) \cdot \Gamma\left(\frac{T + \delta + 1}{2} - \frac{x}{2}\right)}{\Gamma\left(\frac{\delta + 1}{2}\right)^2 \Gamma(T + \delta + 1)}$$

$$\geq \ln \frac{2^{T + \delta}\sqrt{\delta + 1} \cdot \Gamma\left(\frac{T + \delta + 1}{2} + \frac{x}{2}\right) \cdot \Gamma\left(\frac{T + \delta + 1}{2} - \frac{x}{2}\right)}{\pi \Gamma(T + \delta + 1)}$$

$$\geq -\ln(e\sqrt{\pi}) + \frac{1}{2}\ln\left(\frac{1 + \delta}{T + \delta}\right) + \ln\left(\left(1 + \frac{x}{T + \delta}\right)^{\frac{T + \delta + x}{2}}\left(1 + \frac{x}{T + \delta}\right)^{\frac{T + \delta - x}{2}}\right)$$

$$= -\ln(e\sqrt{\pi}) + \frac{1}{2}\ln\left(\frac{1 + \delta}{T + \delta}\right) + (T + \delta)\,\mathrm{D}\left(\frac{1}{2} + \frac{x}{2(T + \delta)} \middle\| \frac{1}{2}\right)$$

$$\geq -\ln(e\sqrt{\pi}) + \frac{1}{2}\ln\left(\frac{1 + \delta}{T + \delta}\right) + \frac{x^2}{2(T + \delta)},$$

where in the first inequality we used Lemma 15, in the second one Lemma 14, and in third one the known lower bound to the divergence $\mathrm{D}\left(\frac{1}{2} + \frac{x}{2} \middle\| \frac{1}{2}\right) \geq \frac{x^2}{2}$. Exponentiating and overapproximating, we get the stated bound. $\qquad \square$

## F.1 Proof of Corollary 5

The Lambert function $W(x) : [0, \infty) \to [0, \infty)$ is defined by the equality

$$x = W(x)\exp\left(W(x)\right) \qquad \text{for } x \geq 0. \tag{21}$$

The following lemma provides bounds on $W(x)$.

**Lemma 17.** *The Lambert function satisfies* $0.6321 \log(x+1) \leq W(x) \leq \log(x+1)$ *for* $x \geq 0$.

*Proof.* The inequalities are satisfied for $x = 0$, hence we in the following we assume $x > 0$. We first prove the lower bound. From (21) we have

$$W(x) = \log\left(\frac{x}{W(x)}\right) . \tag{22}$$

From the first equality, using the elementary inequality $\ln(x) \leq \frac{a}{e} x^{\frac{1}{a}}$ for any $a > 0$, we get

$$W(x) \leq \frac{1}{a\,e}\left(\frac{x}{W(x)}\right)^a \quad \forall a > 0,$$

that is

$$W(x) \leq \left(\frac{1}{a\,e}\right)^{\frac{1}{1+a}} x^{\frac{a}{1+a}} \quad \forall a > 0. \tag{23}$$

Using (23) in (22), we have

$$W(x) \geq \log\left(\frac{x}{\left(\frac{1}{a\,e}\right)^{\frac{1}{1+a}} x^{\frac{a}{1+a}}}\right) = \frac{1}{1+a} \log(a\,e\,x) \quad \forall a > 0 .$$

Consider now the function $g(x) = \frac{x}{x+1} - \frac{b}{\log(1+b)(b+1)} \log(x+1), x \geq b$. This function has a maximum in $x^* = (1 + \frac{1}{b}) \log(1+b) - 1$, the derivative is positive in $[0, x^*]$ and negative in $[x^*, b]$. Hence the minimum is in $x = 0$ and in $x = b$, where it is equal to 0. Using the property just proved on $g$, setting $a = \frac{1}{x}$, we have

$$W(x) \geq \frac{x}{x+1} \geq \frac{b}{\log(1+b)(b+1)} \log(x+1) \quad \forall x \leq b .$$

For $x > b$, setting $a = \frac{x+1}{ex}$, we have

$$W(x) \geq \frac{e\,x}{(e+1)x+1} \log(x+1) \geq \frac{e\,b}{(e+1)b+1} \log(x+1) \tag{24}$$

Hence, we set $b$ such that

$$\frac{e\,b}{(e+1)b+1} = \frac{b}{\log(1+b)(b+1)}$$

Numerically, $b = 1.71825...$, so

$$W(x) \geq 0.6321 \log(x+1) .$$

For the upper bound, we use Theorem 2.3 in [13], that says that

$$W(x) \leq \log \frac{x+C}{1+\log(C)}, \quad \forall x > -\frac{1}{e},\ C > \frac{1}{e}.$$

Setting $C = 1$, we obtain the stated bound. $\qquad\square$

**Lemma 18.** *Define* $f(x) = \beta \exp \frac{x^2}{2\alpha}$, *for* $\alpha, \beta > 0$, $x \geq 0$. *Then*

$$f^*(y) = y\sqrt{\alpha W\left(\frac{\alpha y^2}{\beta^2}\right)} - \beta \exp\left(\frac{W\left(\frac{\alpha y^2}{\beta^2}\right)}{2}\right) .$$

*Moreover*

$$f^*(y) \leq y\sqrt{\alpha \log\left(\frac{\alpha y^2}{\beta^2} + 1\right)} - \beta.$$

*Proof.* From the definition of Fenchel dual, we have

$$f^*(y) = \max_x \; x\,y - f(x) = \max_x \; x\,y - \beta \exp\frac{x^2}{2\alpha} \le x^* \, y - \beta$$

where $x^* = \arg\max_x x\,y - f(x)$. We now use the fact that $x^*$ satisfies $y = f'(x^*)$, to have

$$x^* = \sqrt{\alpha W\left(\frac{\alpha y^2}{\beta^2}\right)},$$

where $W(\cdot)$ is the Lambert function. Using Lemma 17, we obtain the stated bound. $\square$

*Proof of Corollary 5.* Notice that the KT potential can be written as

$$F_t(x) = \epsilon \cdot \frac{2^t \cdot \Gamma(1)\Gamma\left(\frac{t+1}{2} + \frac{x}{2}\right) \cdot \Gamma\left(\frac{t+1}{2} - \frac{x}{2}\right)}{\Gamma(\frac{1}{2})^2\Gamma(t+1)}\,.$$

Using Lemma 16 with $\delta = 0$ we can lower bound $F_t(x)$ with

$$H_t(x) = \epsilon \cdot \exp\left(\frac{x^2}{2t} + \frac{1}{2}\ln\left(\frac{1}{t}\right) - \ln(e\sqrt{\pi})\right)\,.$$

Since $H_t(x) \le F_t(x)$, we have $F_t^*(x) \le H_t^*(x)$. Using Lemma 18, we have

$$\forall u \in \mathcal{H} \qquad F_T^*\left(\|u\|\right) \le H_T^*\left(\|u\|\right) \le \sqrt{T\log\left(\frac{24T^2\|u\|^2}{\epsilon^2} + 1\right)} + \epsilon\left(1 - \frac{1}{e\sqrt{\pi T}}\right).$$

An application of Theorem 3 completes the proof. $\square$

## F.2 Proof of Corollary 6

*Proof.* Let

$$F_t(x) = \frac{2^t \cdot \Gamma(\delta+1)\Gamma(\frac{t+\delta+1}{2} + \frac{x}{2})\Gamma(\frac{t+\delta+1}{2} - \frac{x}{2})}{\Gamma(\frac{\delta+1}{2})^2\Gamma(t+\delta+1)}\,,$$

$$H_t(x) = \exp\left(\frac{x^2}{2(t+\delta)} + \frac{1}{2}\ln\left(\frac{1+\delta}{t+\delta}\right) - \ln(e\sqrt{\pi})\right)\,.$$

Let $f_t(x) = \ln(F_t(x))$ and $h_t(x) = \ln(H_t(x))$. By Lemma 16, $H_t(x) \le F_t(x)$ and therefore $f_t^{-1}(x) \le h_t^{-1}(x)$ for all $x \ge 0$. Theorem 4 implies that

$$\forall u \in \Delta_t \qquad \text{Regret}_t(u) \le f_t^{-1}(\mathrm{D}\left(u\|\pi\right)) \le h_t^{-1}(\mathrm{D}\left(u\|\pi\right))\,.$$

Setting $t = T$ and $\delta = T/2$, and overapproximating $h_t^{-1}(\mathrm{D}\left(u\|\pi\right))$ we get the stated bound. $\square$