[Reviews · NeurIPS 2016]

Reviewer 1

Summary

This paper proposes a new interpretation of parameter-free algorithms by showing the characteristic of these parameter-free algorithms to be coin-betting with maximal reward. It starts from using a betting algorithm based on the Krichevsky-Trofimov estimator with coin-betting potential function, and then generalizes it to online learning optimization over Hibert Space and Learning with expert advices. Real-world datasets also demonstrate the effectiveness of proposed algorithm.

Qualitative Assessment

This paper provides novel understanding of parameter-free algorithms as coin-betting with maximal reward. Based on their understanding, several new algorithms for online convex optimization and learning with experts. The whole paper is well organized and readers can easily follow this paper. Overall, this is a complete work with novel interpretations.

Confidence in this Review

2-Confident (read it all; understood it all reasonably well)


Reviewer 2

Summary

The present paper addresses the online convex optimization setting with a decision set equal to either (i) a Hilbert space or (ii) the probability simplex. The authors present a novel reduction from this problem to a coin betting problem for which optimal and simple algorithms are well known. In particular they show that the Krichevsky-Trofimov strategy (or any generalization of it), which is optimal for coin betting, leads to parameter-free algorithms in (i) and (ii). "Parameter-free" means that the learning algorithm is not tuned as a function of the unknown norm of the competitive vector (for (i)) or the unknown KL divergence of the a good weight vector w.r.t. the the prior at hand (for (ii)). This general framework helps generalize previous works on parameter-free algorithms in either (i) or (ii).

Qualitative Assessment

My opinion about the paper is very good. It is technically solid, very well written, and it helps generalize existing algorithms for Online Linear Optimization (OCO) or Learning with Expert Advice (LEA) through a novel reduction to a coin betting problem. It should be of great interest to people in the online learning community. I must admit that I do not find the coin betting reduction intuitive (especially for LEA, cf. Section 4.2). However, the generality that this approach provides and the simplicity of the resulting algorithms makes it quite attractive. "Parameter-free": you assume that all gains are bounded by 1 (in Euclidean norm for OLO or in sup norm for LEA). Imagine for a second that all gains were bounded by b instead, with an unknown range b. Can you please answer the following? 1. Of course we could rescale all gains by b and apply your machinery to the rescaled gains to get regret bounds that would be proportional to b. However I have the impression that the resulting outputs w_t would depend on the unknown value of b. Is it correct? 2. Fortunately, for the particular case of the Krichevsky-Trofimov potential, the w_t are proportional to the gains. Therefore, in the LEA framework, the resulting weight vectors p_t defined by (12) would not depend on the value of b. Am I correct? 3. Unfortunately it seems that this invariance property does not hold in the OLO framework. Is it true? 4. What happens if we play with losses instead of gains? Again, if all losses are bounded by b, I have the impression that the transformation gain <- 1-loss/b works for LEA (i.e., the algorithm does not depend on b), but we may have a problem for OLO. Is it indeed the case? Please add a discussion of all the four points above in the paper. In particular if I am not wrong for 3. or 4., then the authors should be careful when using the word "parameter-free". Other (minor) comments: - l.16: *the* Hilbert space --> a Hilbert space - Lemma 1: is equivalent to: implies? I do not understand the equivalence. - Definition 2: can you give examples of potentials at this point? (As such, the reader has to wait a little bit.) - l.193, Gamma function: t^{x-1} instead of t^{-x} - end of l.394: the cases u_i=0 or \pi_i=0 should be treated separately (but the result is ok).

Confidence in this Review

3-Expert (read the paper in detail, know the area, quite certain of my opinion)


Reviewer 3

Summary

The paper presents a new intuitive framework to design parameter-free algorithms based on a reduction to betting on outcomes of an adversarial coin. The method shows a new interpretation of parameter-free algorithms as coin-betting algorithms, which reveals the common hidden structure of previous parameter-free algorithms and also allows the design of new algorithms. The paper has also run an empirical evaluation to demonstrate the theory.

Qualitative Assessment

strengths of the paper: 1. The paper has proposed a novel framework to design parameter-free algorithms. The proposed method is simple with no parameters to be tuned. 2. They have both demonstrated in theory and experiments that the proposed algorithm improve or match previous results in terms of regret guarantee and per-round complexity.

Confidence in this Review

1-Less confident (might not have understood significant parts)


Reviewer 4

Summary

In this paper, the authors provide a new approach for solving the classical problem of learning with expert advice (LEA) as well as the problem of online linear optimization (OLO) over Hilbert space. The idea is to reduce these two problem to another classical problem, the coin-betting problem. Then based on an existing coin-betting protocol, the authors are able to obtain parameter-free algorithms for these two problems. The new algorithm for the LEA problem achieves a smaller regret than previous ones, while the new algorithm for the OLO problem matches the regret bound of an existing one.

Qualitative Assessment

There seems to be a mistake in the proof of Lemma 14, so we are not sure about the correctness of Corollaries 5&6. More precisely, the second inequality below line 430 does not hold: it should be \leq instead \geq because \ln(1+x) \leq x-(1-\ln(2))x^2. Our guess is that Lemma 14 is still correct, but the proof needs to be rewritten. Assuming that the bug can indeed be fixed, we find the paper interesting since it provides a new approach for designing online algorithms. This is achieved by establishing a somewhat unexpected connection to another classical problem, the coin-betting problem, which has a long history of works itself. With the help of such a connection, the authors are able to convert an existing coin-betting protocol into new algorithms for the LEA problem and the OLO problem. The resulting algorithms are parameter free, which means that no parameter tuning is needed, and this adds to the strength of this paper. In addition, the paper is well-written and easy to follow in general. Two possible weakness we find are the following. First, the potential function used by this paper seems rather complex and unintuitive, which may limit its impact. It is not clear if it is easy to design different potential functions to yield different online algorithms, just as what the mirror-descent algorithm can provide. Next, the OLO problem for which the new algorithm works is the unconstrained version, instead of the more popular constrained one. It is not clear if the ideas developed in this paper can be used to solve the more general OLO problem over any convex feasible set.

Confidence in this Review

2-Confident (read it all; understood it all reasonably well)


Reviewer 5

Summary

Many online learning algorithms are based on a learning parameter which controls how far the algorithm changes its answer after receiving new data. While many of these algorithms fix this rate in advance, others change it as they progress. The main problem discussed in this paper is trying to construct an algorithm that is comparable to the case where we know in advance all the data and not receive it part by part. In this paper, the authors construct a strategy for a coin betting game such that in each turn the gambler needs to decide how much to bet and on which outcome. This strategy is based on suitably chosen potential functions. Each such family of functions produce a betting strategy which also gives an upper bound on the regret for the game, i.e. how much money the gambler could have won if he knew all the coin tosses in advance compared to how much he won using this strategy. The authors then use this coin betting strategy to solve other online learning problems such as the Online Learning Optimization and Learning with Expert Advice problems. This strategy seems to generalize other parameter-free algorithms which already exist.

Qualitative Assessment

The coin betting game and both the Online learning optimization and Learning with expert advice are interesting problems in online learning, and the choice of the learning rate is fundamental to many learning algorithms. Thus any framework that provides a way to chose this rate in a smart way is highly interesting and can have many applications. The main issue with the paper is the definition of Coin betting potential which lead to the choice of the learning rate. While the formula below line 137 explains why the condition in line 130 is useful, and lines 138-140 explain from where the expression for beta_t comes from (though it is a bit confusing explanation), it is not at all clear how to construct such a function. Later on it is shown that the known online learning algorithms have corresponding potential functions. It would be really interesting if the authors can give some intuition on how to produce more such functions, thus saying that not only their algorithm generalizes the known results, but actually can produce new ones (and not just other similar potentials). some other minor comments: line 16 - "the Hilbert space" should be "a Hilbert space" unless this space is defined before. line 24-29 - It might be better to add some explanation what is a learning rate (and not just how you use it in the examples of OGD). If the reader didn't know what learning rate is before reading this paper, he would just be more confused after this introduction. Also, it might be a good idea to write that "parameter free" means that we do not set the learning rate in advance. It can also be as simple as writing "the learning parameter" or the "learning rate or leaning parameter" instead of just the "learning rate". line 49 - what do you mean by dom(f) not empty? Is the domain not all of V? if this is possible, then you should write it explicitly. line 109 - The inequality in (5) doesn't seem trivial. If it is just some calculations, maybe add another step or say that this is the case. line 114 - why are the names of Krichevsky and Trofimov in Blue? line 141 - The coin betting seems to be one dimensional always. What is the meaning of infinite dimensional in this line? line 196 - is this "peculiar property" has any significance ?

Confidence in this Review

1-Less confident (might not have understood significant parts)